# ImpressLearn: Continual Learning via Combined Task Impressions

## Abstract

This work proposes a new method to sequentially train deep neural networks on multiple tasks without suffering catastrophic forgetting, while endowing it with the capability to quickly adapt to unseen tasks. Starting from existing work on network masking (Wortsman et al., 2020), we show that simply learning a linear combination of a small number of task-specific supermasks (*impressions*) on a randomly initialized backbone network is sufficient to both retain accuracy on previously learned tasks, as well as achieve high accuracy on unseen tasks. In contrast to previous methods, we do not require to generate dedicated masks or contexts for each new task, instead leveraging transfer learning to keep per-task parameter overhead small. Our work illustrates the power of linearly combining individual impressions, each of which fares poorly in isolation, to achieve performance comparable to a dedicated mask. Moreover, even repeated impressions from the same task (homogeneous masks), when combined, can approach the performance of heterogeneous combinations if sufficiently many impressions are used. Our approach scales more efficiently than existing methods, often requiring orders of magnitude fewer parameters and can function without modification even when task identity is missing. In addition, in the setting where task labels are not given at inference, our algorithm gives an often favorable alternative to the one-shot procedure used by Wortsman et al. (2020). We evaluate our method on a number of well-known image classification datasets and network architectures.

## 1 Introduction

Sequential learning without catastrophic forgetting has been an area of active research in machine learning for some time (Maes et al., 1996; Thrun & Pratt, 1998; Serra et al., 2018). A precondition for achieving Artificial General Intelligence is that models should be able to learn and remember a wide variety of tasks sequentially, without forgetting previously learned ones. In real-world scenarios, data from different tasks may not be available simultaneously, which makes it imperative to both allow continued learning of a potentially unbounded number of tasks (see also *The Sequential Learning Problem* (McCloskey & Cohen, 1989), *Constraints Imposed by Learning and Forgetting Functions* (Ratcliff, 1990) and *Lifelong Learning Algorithms* (Thrun & Pratt, 1998)). Recently, some successful approaches to combat this problem use task specific sub-models, which allow neural networks to context-switch between different learning tasks (Wortsman et al., 2020; Mallya et al., 2018; Mancini et al., 2018). The underlying context for each task can be represented as "*filters*" or "*masks*", altering the network's structure for each task. Yet all of these approaches scale unfavorably with the number of unique tasks to be learned.

**ImpressLearn.** We propose a novel method leveraging transfer learning and network masking to sequentially learn a practically unlimited number of tasks with much lower per-task parameter overhead compared to prevailing benchmarks. Our method, termed *ImpressLearn*, uses elements from *Supermasks in Superposition (SupSup)* by Wortsman et al. (2020); SupSup uses the observation that randomly initialized neural networks contain subnetworks, obtained by applying binary parameter masks (supermasks), that achieve good performance on any particular task (Zhou et al., 2019). These supermasks can be learned with standard gradient descent and stored, one mask per task. At inference, the appropriate task-specific mask is

applied when task identity is known. When task-labels of a previously seen task are unavailable, the correct mask can be inferred via entropy minimization.

While SupSup provides strong results, it restricts knowledge transfer between tasks and requires considerable memory overhead for each additional task. ImpressLearn, on the other hand, solves both problems with one stone: it accommodates positive knowledge transfer by reusing existing supermasks and, as a result, requires much less additional parameters for each new task. Our supermasks (*basis-masks*) are constructed from a small batch of initial *basis-tasks* (heterogeneous setting), or from just a single basis-task (homogeneous setting). By default, we select the first $N$ tasks as basis-tasks where $N$ controls the performance-cost trade-off. Leveraging transfer learning, this set of learned basis-masks, each of which can be interpreted as an impression of a previously seen basis-task, serves as a collection of latent features for new learning objectives, encoding common structural information. This might be reminiscent of associative learning where impressions of previous scenarios are combined to cope with new ones. Once a set of basis-masks is identified, we learn an appropriate linear combination of these impressions to quickly construct a real-valued mask that performs well on an unseen task. Hence, apart from a fixed number of basis-masks, only a small number of floating-pint coefficients need to be learned and stored for each subsequent task. This greatly benefits scalability, a major drawback of previous methods (Wortsman et al., 2020; Mallya et al., 2018). In principle, the efficiency of ImpressLearn allows for an unlimited number of new tasks. In particular, as the number of tasks increases, the per-task memory overhead converges to just a few extra parameters after amortizing the cost of basis-masks, which is considerably cheaper than allocating additional parameter masks for each task.

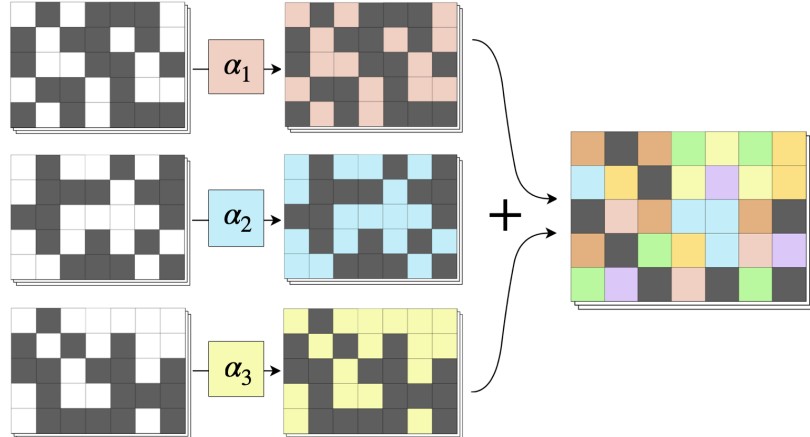

Figure 1: Intuitive representation of ImpressLearn. Binary basis-masks (left) are linearly combined via learnable coefficients $\alpha$ to construct a task-specific real-valued mask (right) that is applied to a fixed randomly initialized backbone network at inference.

**Homogeneous and random basis-masks.** Somewhat surprisingly, we can even generate all basis-masks from the same initial task using different random seeds for the learning algorithm (but the same randomly initialized backbone network). We show that with a sufficiently large number of such homogeneous impressions, our algorithm learns linear combinations with close to benchmark accuracy on new tasks. We are reminded of an infant learning by taking different "snapshots" of the same object to infer properties of another. This homogeneous setting is particularly useful to address possible drifts in the data; akin to ensembling, it leverages the power of linear combinations for transfer learning. An additional important advantage is that the homogeneous setting has no limit on the number of basis-masks we can generate ab-initio.

As another baseline for ImpressLearn, we experimented with optimizing for a linear combination of entirely random masks of desired sparsity. We demonstrate on several benchmarks that if we choose a sufficiently large collection of such random basis-masks, our optimization still yields competitive performance. While combinations of random masks naturally lag behind the heterogeneous and the homogeneous settings, we show that there is a trade-off between the number of masks and their task-specificity (non-randomness). In settings where producing task-specific basis-masks is costly, optimizing for a linear combination of a large number of random masks can still yield satisfactory results.

**Example: LeNet-300-100 on RotatedMNIST.** As a preview of our approach and its performance, Figure 2 shows the accuracy of ImpressLearn compared to SupSup on RotatedMNIST dataset. First, as a sanity check, we apply basis-masks obtained from one task to tasks they were not optimized for. As expected, this yields essentially random accuracy (see X in Figure 2), confirming that the performance of ImpressLearn is beyond pure transfer learning and comes from linearly combining the initial impressions. Next, Figure 2 illustrates that ImpressLearn even with a small number of heterogeneous basis-masks is on par or even superior to SupSup on unseen tasks. Note that, for each additional non-basis task, ImpressLearn with 10 basis-masks requires only $3 \times 10 = 30$ parameters (one per basis-mask per layer); in contrast, SupSup needs to generate an entire binary mask, which requires $25,000+$ tensor indices to specify assuming $10\%$ sparsity. Figure 2 also illustrates the performance of ImpressLearn over homogeneous basis-masks; in this setting, we need a larger number of basis-masks to achieve accuracy comparable to the heterogeneous scenario. Ultimately, however, we still match the performance of SupSup with a vastly smaller number of additional per-task parameters. Lastly, the rightmost plot in Figure 2 shows that our algorithm can successfully operate when task identity is not provided at inference (cf. GN regime (Wortsman et al., 2020)). Here, our optimization procedure with the entropy objective finds the "correct" basis-mask or a linear combination of basis-masks yielding similar or better performance compared to the one-shot baseline from Wortsman et al. (2020). In Section 4, we provide empirical evidence of the efficacy of ImpressLearn on a variety of benchmarks, outlining the radical savings in parameters that need to be stored per-task compared to SupSup.

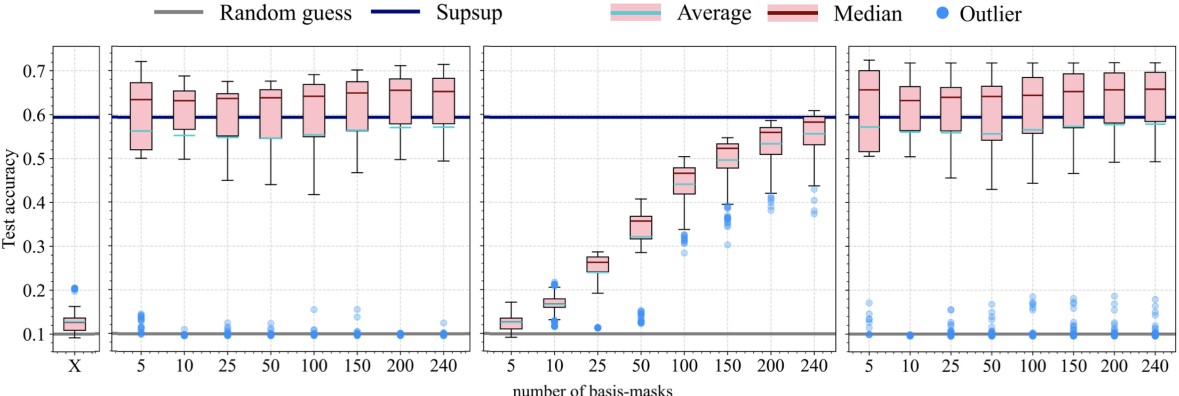

Figure 2: Average test performance (RotatedMNIST) of various strategies over the number of basis-masks. Boxplots from left to right: (1) A single basis-mask X on 10 unseen tasks, (2) ImpressLearn with heterogeneous masks on 10 unseen tasks, (3) ImpressLearn with homogeneous masks on 10 unseen tasks, (4) ImpressLearn with heterogeneous masks on basis-tasks in the GN regime (no task identity available at inference). All results are averaged over 3 different seeds and over masks of sparsity from $\{0.95, 0.92, 0.9, 0.85, 0.8\}$

**The GN regime.** In principle, both SupSup and ImpressLearn require access to task identities in order to apply the right mask to the backbone network during inference. To relax this requirement, Wortsman et al. (2020) extend their algorithm to a more challenging regime (coined *GN*—Given/Not given) where task identifiers are present during training but unavailable at inference. In this regime, Wortsman et al. (2020) use a one-shot minimization of the entropy of the model's outputs to single out the correct mask. In Section 4, we show that our algorithm is able to achieve this feat, too. We demonstrate that, when applied to basis-tasks in the GN regime, the optimization routine of ImpressLearn either identifies the corresponding basis-mask or even finds a better performing combination of basis-masks.

The remainder of the paper is organized as follows. In Section 2, we briefly review related work and general approaches to countering catastrophic forgetting, highlighting research that motivated our approach. In Section 3, we detail the ImpressLearn algorithm as well as discuss its components and features. In Section 4, we demonstrate the effectiveness of ImpressLearn on a variety of datasets and architectures to show close-to-benchmark performance with a drastically reduced parameter count on unseen tasks, especially where the number of incoming tasks is large. Additionally, we conduct several ablation studies to provide

better understanding of the algorithm. Finally, in Section 5, we talk about limitations of our work and avenues for future research.

## 2 Related Work

In practice, intelligence systems should be capable of learning a variety of tasks incrementally and without experiencing catastrophic forgetting—degrading performance on previously acquired skills (McCloskey & Cohen, 1989), while ideally transferring current knowledge to facilitate future training. Generally, tasks are not available concurrently, prohibiting joint training. Conversely, ordinary finetuning (continued training of a pretrained network) inevitably leads to catastrophic forgetting. Continual learning encompasses a broad spectrum of algorithms and architectures that address these issues and propose systems capable of learning from an incremental stream of tasks while minimizing catastrophic forgetting. Most naturally, these techniques are categorized into three groups described below (Delange et al., 2021; Wortsman et al., 2020).

**Regularization-based methods.** The algorithms in this class solve new tasks by finetuning parameters optimized for previous tasks and leverage regularization to combat forgetting. A number of studies assess the importance of individual parameters for previous tasks and penalize their displacement accordingly during optimization. Pioneering this approach, Serra et al. (2018) estimate parameter importance using a Laplace-approximated posterior distribution after training on earlier tasks. Zenke et al. (2017) impose a quadratic penalty proportional to the accumulated sensitivity of previous loss functions to perturbations in the corresponding parameter; Aljundi et al. (2018) use the same strategy but accumulate sensitivity of the network output to parameter perturbations instead. In contrast, Li & Hoiem (2018) regularize by means of distilling current knowledge on the incoming task's data and using it during finetuning. Regularization-based methods require no additional per-task memory overhead and hence are advantageous in capacity-constrained settings. However, the plasticity of a network decreases with more tasks, imposing a natural limit on new tasks, which creates a trade-off between learning new tasks and forgetting.

**Replay methods.** Techniques in this category preserve performance on prior tasks by replaying, rehearsing or otherwise utilizing representative samples from the corresponding data distributions. Most commonly, replay models store examples of seen data in a separate memory buffer (Rebuffi et al., 2017; Rolnick et al., 2019; Riemer et al., 2019); others maintain generators that approximate the original data distribution and provide pseudo-examples (Atkinson et al., 2021; Shin et al., 2017). While the majority of algorithms in this group replay stored examples during optimization to mimic joint training, Lopez-Paz & Ranzato (2017) use them to constrain optimization space and ensure positive knowledge transfer. Replay methods require additional memory to store data samples or allocate generators, however, these costs are usually kept fixed. For this reason, like regularization-based models, replay models exhibit poor stability-plasticity trade-off with more tasks and naturally come with increased memory requirements when compared to regularization-based methods.

**Parameter isolation methods.** Most methods in this category allocate new parameters for incoming tasks and feature little to no interference between previously learned tasks. Rusu et al. (2016) allocate a new copy of the network and enable forward transfer learning with lateral connections going into new modules. Ren et al. (2017) combine individual learners in a decision tree and eliminate outdated models using tree pruning. Another recent publication learns different tasks within separate orthogonal subspaces of the parameter space of a single model, avoiding interference between tasks (Cheung et al., 2019). A large body of recent algorithms piggyback on a single backbone network shared by all tasks. As such, Wen et al. (2020) (BatchEnsemble) operate on a fixed pretrained network and, for each incoming task, optimize for a rank-one parameter mask applied to the backbone at inference. Mallya & Lazebnik (2018) (PackNet) use pruning to assign subsets of free parameters of a backbone network to individual tasks by issuing one binary parameter mask per task. These assigned parameters are forever frozen at their trained values, limiting the capacity of the network for future tasks. In a subsequent study, Mallya et al. (2018) (Piggyback) lift this limitation by directly optimizing per-task binary masks and applying them to a fixed pretrained network.

**Supermasks in Superposition (SupSup).** ImpressLearn is most closely related to yet another similar method called SupSup (Wortsman et al., 2020). This algorithm trains individual per-task binary masks and applies them to a fixed randomly-initialized network, leveraging the existence of supermasks (Zhou et al., 2019). The mask optimization algorithm, edge-popup (Ramanujan et al., 2019), uses a heaviside function to binarize mask values on the forward pass and employs a straight-through estimator when computing gradients. In addition, Wortsman et al. (2020) propose different training and inference modes depending on availability of task identifiers; e.g., GG refers to the scenario when task identifies are known during both training and inference, while in GN they are available only during training. In the latter case, Wortsman et al. (2020) introduce a one-shot algorithm to infer task identity by minimizing entropy of the output starting from a uniform linear combination of masks and optimizing the coefficients. While SupSup and Piggyback suffer no catastrophic forgetting as there is naturally no interference between tasks, these algorithms allow no knowledge transfer for the exact same reason. This independence on previous learning comes at a cost of significant memory overhead associated with solving each additional task. ImpressLearn fixes both issues by recycling parameter masks dedicated for a fixed set of previous tasks while learning new ones.

## 3 Approach

**Preliminaries and notation.** We largely adopt the notation from Wortsman et al. (2020). Given an $l$-way classification task $t \in T$ with dataset $(X^t, Y^t)$ from a stream of tasks $T$, let $f(W, \cdot)$ be a $d$-layer neural network parameterized by randomly initialized weights $W = \{W_i\}_{i=1}^d$. Following Wortsman et al. (2020), we use the edge-popup algorithm (Ramanujan et al., 2019) to train a binary parameter basis-mask $M^t = \{M_i^t\}_{i=1}^d$ for each basis-task $t \in T$. The sparsity of a mask $M^t$ is given by the fraction of zeroes in it.

**The ImpressLearn algorithm.** We define a set of basis-tasks $T_b \subset T$. By default, basis-tasks are the first $N$ tasks encountered in the stream $T$ where $N$ depends on the amount of available memory; in our experiments, we test ImpressLearn across different values of $N$ ranging from 3 to 240 (Section 4). The backbone network is randomly initialized using Kaiming normal distribution and remains fixed at all times (He et al., 2020). Then, the algorithm proceeds as follows:

1. For a basis-task $t \in T_b$ we use the edge-popup algorithm to create a basis-mask $M^t$, leading to a collection $\mathcal{M} = \{M^1, M^2, \ldots M^N\}$ of basis-masks.

2. For a new task $s \in T \setminus T_b$ we define a matrix $\alpha^s \in \mathbb{R}^{N \times d}$ of learnable coefficients, one per basis-mask per layer. The linear combination of basis-masks solving $t$ is found by SGD applied to

$$\hat{\alpha}^s = \arg\min_{\alpha^s} \mathcal{L}\left(f\left(W \odot \sum_{t=1}^N \alpha_t^s M^t, X^s\right), Y^s\right) \tag{1}$$

where $\mathcal{L}$ is the cross-entropy loss function.

**Mask generation & edge-popup.** ImpressLearn uses the edge-popup algorithm developed by Ramanujan et al. (2019) to generate basis-masks. This method introduces learnable "popup scores" $s_i$ associated with each prunable parameters $w_i$ in the immutable backbone network. During optimization, a binary thresholding function determines if a particular weight participates in the forward pass based on its popup score, while a straight-through gradient estimator is employed on the backward pass. The desired sparsity level is set beforehand and is controlled by the threshold value. In principle, ImpressLearn can use other methods for mask generation, e.g., iterative magnitude pruning (Frankle & Carbin, 2019; Zhou et al., 2019).

**Initialization of $\alpha$.** A priori all basis-masks have equal chance to contribute to solving the new task. Moreover, testing any single basis-mask optimized for task $t$ on any other task $t' \neq t$ gives almost random performance, implying that there is no direct knowledge transfer (see Section 4). Hence, a uniform prior on coefficients $\alpha$ is a reasonable assumption. We treat each layer independently, setting $\alpha_{t,i} = 1/N$ for all $i \in [d]$ at the start of optimization.

**Regularization.** While overfitting is less of a concern given the small number of parameters in our optimization equation (1), it is desirable that ImpressLearn discovers a sparse solution (i.e., uses as few basis-masks as possible). This is especially relevant when performing inference on basis-tasks, where ImpressLearn is expected to identify the "correct" mask among all basis-masks. Hence, we apply regularization on parameters $\alpha$ for each layer to obtain the regularized loss function

$$J = \mathcal{L}\left(f\left(W \odot \sum_{t=1}^{N} \alpha_t^s M^t, X^s\right), Y^s\right) + \lambda \sum_{i=1}^{d}\left(\sum_{t=1}^{N} |\alpha_{t,i}^s| - 1\right)^2 \tag{2}$$

**Heterogeneous, homogeneous & random masks.** As we discussed above, ImpressLearn creates one basis-mask for each task in the collection of basis-tasks $T_b$ by default, which we call the *heterogeneous* mode. In this setting, we leverage knowledge transfer from all basis-tasks to solve new tasks. However, when tasks are scarce, limiting basis-mask generation to one per task affects the viability of our method and does not allow for scaling benefits to become apparent. For example, in the case of Split-CIFAR-100 (20 tasks), we can generate at most 20 heterogeneous basis-masks. Hence, we also evaluate our approach with a set of *homogeneous* basis-masks that all come from the same basis-task. A priori it is unclear whether masks produced on the same backbone network for the same basis-task are sufficiently different to generate a diverse enough basis set. While previous work suggests that wider network architectures can support multiple suitable subnetworks for a given task Ramanujan et al. (2019); Frankle & Carbin (2019), we find this effect is prominent even using relatively conservative architectures such as LeNet-300-100 with PermutedMNIST (Lecun et al. (1998)). Our experiments show that basis-masks are very sensitive to the initialization of popup scores and to data order: the overlap of homogeneous masks produced with different random seeds is close to random. Hence, at sufficiently low sparsities, homogeneous masks are practically independent. Finally, note that optimization of homogeneous basis-masks can be parallelized as it assumes access to only one task, potentially bringing significant computational speedups.

To quantify the importance of task-specific data in mask generation, we experimented with a variation of ImpressLearn where basis-masks are drawn randomly and not optimized (see Section 4 and B). This allows us to study trade-offs between random and task-specific masks and provide an alternative when optimizing basis-masks with edge-popup is too resource-consuming. While in general this approach requires more basis-masks to reach commensurate performance, it offers a competitive alternative when optimization is costly.

## 4   Experimental Results

In this Section, we test how well ImpressLearn generalizes to unseen tasks and retains its performance on the learned ones. To achieve this, we evaluated ImpressLearn on standard continual learning benchmarks: MNIST (Lecun et al., 1998): Permuted and Rotated; Split-CIFAR-100 (Krizhevsky, 2012) and Split-ImageNet (Deng et al., 2009). A detailed description of our experimental choices and infrastructure is given in Appendix A.

**Main results.** Overall, our experiments confirm that ImpressLearn is capable of successfully learning new tasks with a fraction of the parameters required by other approaches. In line with expectations, we see a strong positive relationship between accuracy and the size $N$ of the impression set $\mathcal{M}$, saturating at different data-specific sizes. While the number of heterogeneous masks required for competitive accuracy varies by dataset, ImpressLearn is particularly resource-efficient when the number of possible tasks is large so that it becomes prohibitively costly to store a separate masks for each task (e.g. PermutedMNIST with 784 factorial tasks). On the other hand, when the number of available tasks is limited (e.g., Split-CIFAR-100, Split-ImageNet), ImpressLearn exhibits larger performance gaps compared to SupSup, hinting that not enough basis-masks are generated. Compared to the heterogeneous scenario, we need more homogeneous basis-masks to achieve similar performance (and even more random masks, see B). For SplitImageNet, we only show results for heterogeneous basis-masks with $N \leq 35$ due to limitations on compute (Figure 5). We anticipate that with a larger collection of basis-masks ImpressLearn can match the performance of SupSup. Additionally, note that ImpressLearn's $\alpha$-optimization procedure yields performance superior to SupSup's one-shot entropy minimization under the GN regime on SplitImageNet.

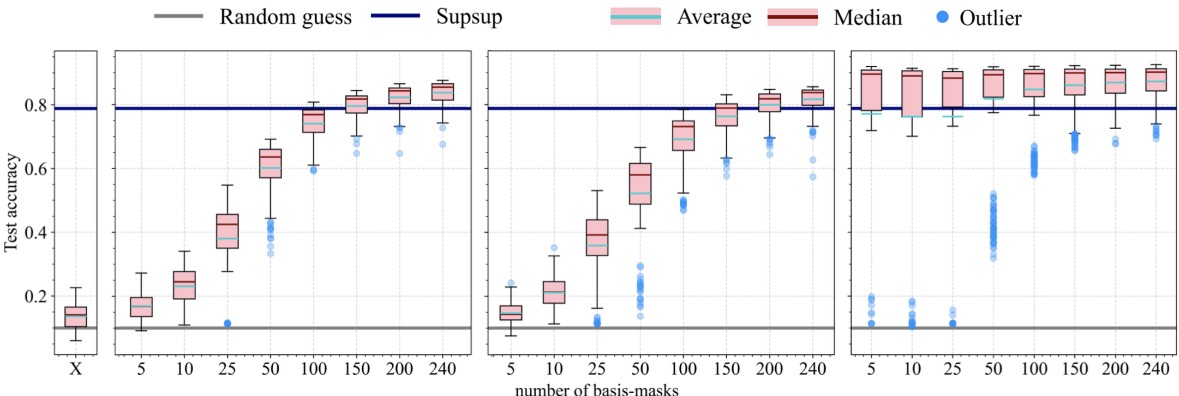

Figure 3: Average test performance (PermutedMNIST) of various strategies over the number of basis-masks. Boxplots from left to right: (1) A single basis-mask X on 10 unseen tasks, (2) ImpressLearn with heterogeneous masks on 10 unseen tasks, (3) ImpressLearn with homogeneous masks on 10 unseen tasks, (4) ImpressLearn with heterogeneous masks on basis-tasks in the GN regime (no task identity available at inference). All results are averaged over 3 different seeds and over masks of sparsity from $\{0.95, 0.92, 0.9, 0.85, 0.8\}$

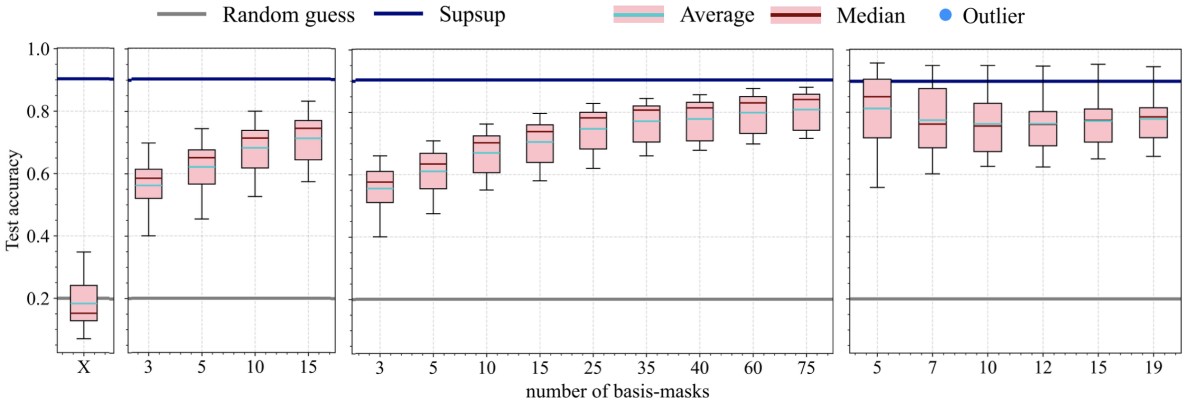

Figure 4: Average test performance (Split-CIFAR-100) of various strategies over the number of basis-masks. Boxplots from left to right: (1) A single basis-mask X on 5 unseen tasks, (2) ImpressLearn with heterogeneous masks on 5 unseen tasks, (3) ImpressLearn with homogeneous masks on 5 unseen tasks, (4) ImpressLearn with heterogeneous masks on basis-tasks in the GN regime (no task identity available at inference). All results are averaged over 3 different seeds and over masks of sparsity from $\{0.95, 0.92, 0.9, 0.85, 0.8\}$

**Ablation study: incorrect masks.** We confirm that all performance of ImpressLearn comes not from chance application of a suitable mask that does well on other tasks but rather from appropriately combining multiple masks through a learned linear function. In particular, we test impressions derived from one basis-task on other tasks and find that the incorrect basis-mask fails to have any predictive power on other tasks when used in isolation, giving a roughly random accuracy of $10 \pm 3\%$ for Permuted/Rotated-MNIST/Split-ImageNet (Figures 2, 3, and 5) and $20 \pm 2\%$ on Split-CIFAR-100 (Figure 4), respectively.

**Ablation study: Hybrid ImpressLearn.** In order to understand the value of linear combination of basis-masks, we implemented a hybrid method that follows ImpressLearn in all hidden layers (i.e., optimizes a linear combination of basis-masks for each new task) and SupSup in the output layer (i.e., allocates a learnable binary parameter-mask for each new task). This hybrid approach is only marginally better than ImpressLearn and only when the number of basis-masks is small. As the number of basis-masks increases, the two methods yield equivalent performance (Section C and Figure 10). This demonstrates that ImpressLearn is not just a simplified version of SupSup but rather a novel algorithm with its unique properties and core

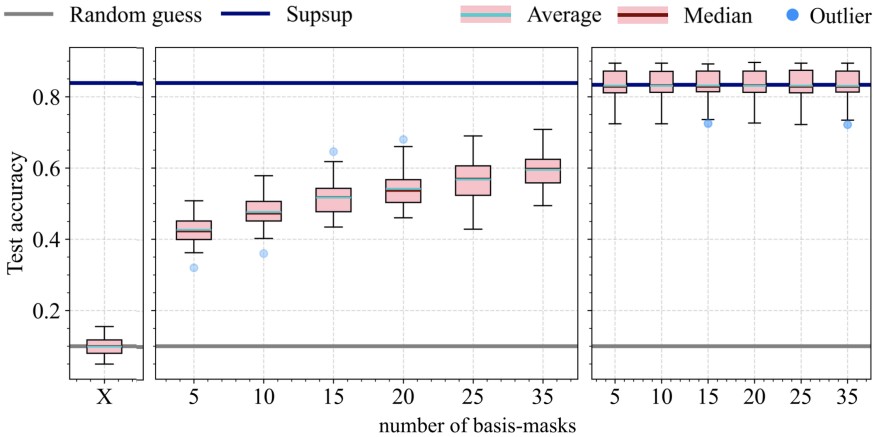

Figure 5: Average test performance (Split-ImageNet) of various strategies over the number of basis-masks. Boxplots from left to right: (1) A single basis-mask X on 5 unseen tasks, (2) ImpressLearn with heterogeneous masks on 5 unseen tasks, (3) ImpressLearn with heterogeneous masks on basis-tasks in the GN regime.

principles. Additionally, these ablation experiments show that ImpressLearn is unlike trivial finetuning of the classification layer; instead, it takes advantage of all layers to reach its performance.

**GN regime.** To evaluate the strength of linearly combining impressions, we studied the regime where task identifiers are not provided at inference (GN). In the first set of experiments, we use ImpressLearn's $\alpha$-optimization with the entropy objective and the one-shot entropy minimization from Wortsman et al. (2020) over a linear combination of all basis-masks with respect to one of the basis-tasks from $T_b$ without explicitly providing its identity. For most architecture-dataset pairs, our strategy either determines the correct basis-mask or discovers an even better-performing linear combination of basis-masks (the right-most plots in Figures 2, 3, 4, and 5), facilitating positive knowledge transfer to previous tasks. On the other hand, SupSup's one-shot procedure outputs only a single binary mask and will retain the original performance at best. Thus, as shown in Figure 6, ImpressLearn applied to unknown basis-tasks outperforms SupSup in both GN and GG regimes, i.e., even when task labels are available to SupSup and not ImpressLearn. Similar results are observed when applying the two algorithms in the GN regime on a mixture of basis-masks and real-valued masks precomputed from the coefficients discovered by ImpressLearn for non-basis tasks. As before, our optimization routine compares favourably to one-shot entropy minimization.

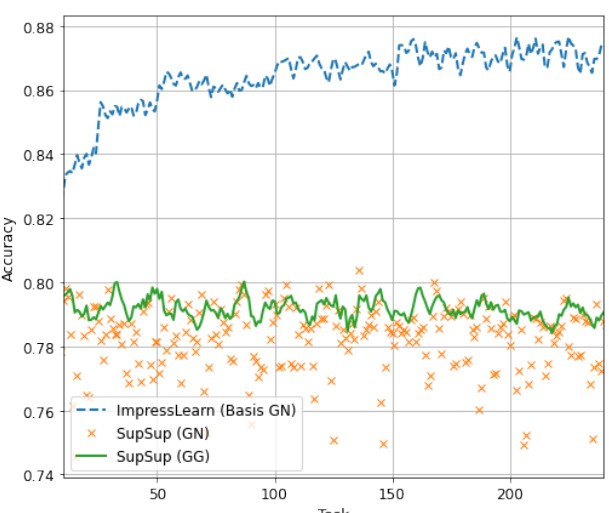

Figure 6: Validation accuracy of ImpressLearn-GN, SupSup-GG, and SupSup-GN (+one-shot entropy minimization) on PermutedMNIST. In the GN regime, results are averaged across 10 different data splits and orders per seed and sparsity. All results are averaged across 3 different seeds and masks of sparsity $\in \{0.95, 0.92, 0.9, 0.85, 0.8\}$.

**Model efficiency & parameter savings.** We present additional information about ImpressLearn in various experimental settings and compare its memory costs with those of SupSup (Table 1). Our approach has a fixed cost of storing basis-masks in addition to minimal per-task costs associated with allocating linear

coefficients ($\alpha$). Amortizing mask storage across all available tasks, ImpressLearn affords impressive savings in memory at the expense of either no or small loss in performance, which can be further mitigated by generating more basis-masks.

Table 1: Dataset, parameter, and memory overhead statistics for models used in this study. The total feasible number of tasks generated from a dataset is presented in the second column. The number of classes per task is denoted by K. Parameter counts do not include biases or batchnorm parameters. Mask size (in kB) is the space on disk required to store each additional parameter mask. The number of basis-masks $N$ was approximately chosen to give a close-to-benchmark performance on new tasks. The number of per-task floating-point parameters required by ImpressLearn is denoted by $|\alpha|$. Cost* (in kB) is the per-task memory required by ImpressLearn with costs of basis-masks amortized across all tasks. Ratio denotes the savings factor when comparing memory costs of ImpressLearn with those of SupSup when all available tasks are learned.

| Dataset | Tasks | K | Model | Params | Mask | $N$ | $|\alpha|$ | Cost* | Ratio |
|---------|-------|---|-------|--------|------|-----|-----------|-------|-------|
| PermutedMNIST | 784! | 10 | LeNet-300-100 | 266K | 65 | 100 | 300 | 1 | 55 |
| RotatedMNIST | 359 | 10 | LeNet-300-100 | 266K | 65 | 5 | 15 | 1 | 67 |
| Split-CIFAR-100 | 20 | 5 | ResNet-18 | 6.2M | 1,500 | 10 | 210 | 750 | 2 |
| Split-ImageNet | 100 | 10 | ResNet-50 | 25.6M | 6,250 | 75 | 3,975 | 240 | 26 |

## 5  Discussion

In this work, we use elements of an existing continual learning algorithm (SupSup) to design our own (ImpressLearn) that, leveraging principles from transfer learning to generalize to new tasks, allows for scalable and parameter-efficient continual learning. By linearly combining supermasks that solve previously seen basis-tasks, ImpressLearn is capable of learning new tasks effectively, allowing positive knowledge transfer in both directions. We show that this effect is consistent across task types and network architectures, and that it achieves competitive performance while using significantly fewer parameters. This work highlights the advantages of reusing existing meta-features learned on previous tasks for future learning problems and opens up a space of possibilities of applying transfer learning to protect against catastrophic forgetting. Future research may explore values of the coefficients $\alpha$ that arise as a result of our optimization, which may provide further insights into the role of individual impressions in task-specific linear combinations. Further, a closer comparison between our $\alpha$-optimization and SupSup's one-shot entropy minimization may shed light on crucial components that empower ImpressLearn.

**Limitations.** Our experimental results show that ImpressLearn works well on several benchmarks, and particularly shines when the number of new tasks is large. In scenarios where the number of different tasks is small (e.g., Split-CIFAR-100), our approach yields slightly inferior performance and can only give limited parameter savings, if any. We hypothesize that, in the case of task streams this short, the natural limit on the number of generated heterogeneous basis-masks prevents reaching the performance of SupSup. For Split-ImageNet, the amount of available compute and resources limited the number of adopted basis-masks; we believe that increasing the size of the impression pool can further boost the performance.

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

## A   Experimental details

Our experiments encompassed a range of sparsities: $\{0.95, 0.92, 0.9, 0.85, 0.8\}$. Each seed varied the random initialization of edge-popup scores and the training data order. In the heterogeneous mode, we used one seed to generate all basis-masks $\mathcal{M}$. In the homogeneous scenario, we seeded every mask to ensure mask diversity. The backbone network was kept fixed for each unique combination of $\mathcal{M}$ and $\alpha$-optimization for new tasks. All runs were performed on three different backbone networks and train/test splits to ensure that results were sufficiently general. The presented boxplots show averages over all settings and sparsity levels. All experiments were performed on an internal cluster with NVIDIA V100 Tesla and RTX 8000 GPUs.

Table 2: Overview of hyperparameters ($\alpha$-optimization). We employed momentum of 0.9 and weight decay of 0.1 for the Adam optimizer.

| Model | Dataset | Learning rate | $\lambda$ | Optimizer | Batch size |
|---|---|---|---|---|---|
| LeNet-300-100 | RotatedMNIST | 0.002 | 0 | RMSprop | 128 |
| LeNet-300-100 | PermutedMNIST | 0.002 | 0 | RMSprop | 128 |
| ResNet-18 | Split-CIFAR-100 | 0.02 | 0.005 | Adam | 64 |
| ResNet-50 | Split-ImageNet | 0.0025 | 0.005 | Adam | 96 |

## B    Random Basis-masks

We compare performances of random basis-masks and homogeneous masks to illustrate the trade-offs when basis-mask optimization with edge-popup is prohibitively expensive (Figures 7, 8 and 9). Generally, as the number of random masks increases, this modification of ImpressLearn fares surprisingly well and could become an alternative for resource-constrained scenarios with only a slight degradation in accuracy.

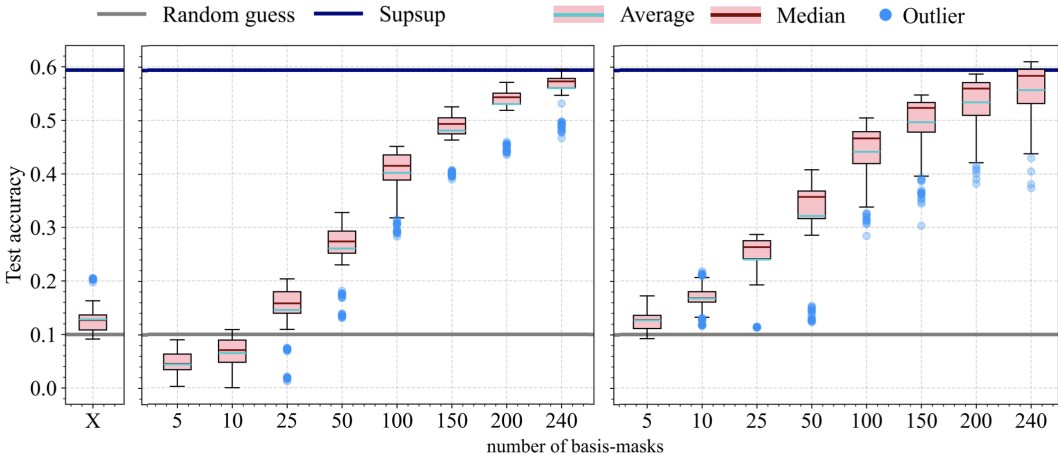

Figure 7: Average test performance (RotatedMNIST) of various strategies on 10 unseen tasks over the number of basis-masks. Boxplots from left to right: (1) A single basis-mask X, (2) ImpressLearn with random masks, (3) ImpressLearn with homogeneous masks. All results are averaged over 3 different seeds and over masks of sparsity from $\{0.95, 0.92, 0.9, 0.85, 0.8\}$

## C    Hybrid ImpressLearn

As another ablation experiment, we employ ImpressLearn for all hidden layers of the model but learn a separate binary parameter mask for the output layer with edge-popup for each incoming task, just as SupSup does. As the number of basis impressions grows, the difference between this hybrid approach and our ImpressLearn vanishes (Figure 10). Hence, allowing more parameters in the last layer (for an additional edge-popup mask) does not improve the performance and the power of ImpressLearn does not reside solely in finetuning parameters for new tasks. Lastly, these results evidence that ImpressLearn is qualitatively different from SupSup and offers its own benefits and trade-offs.

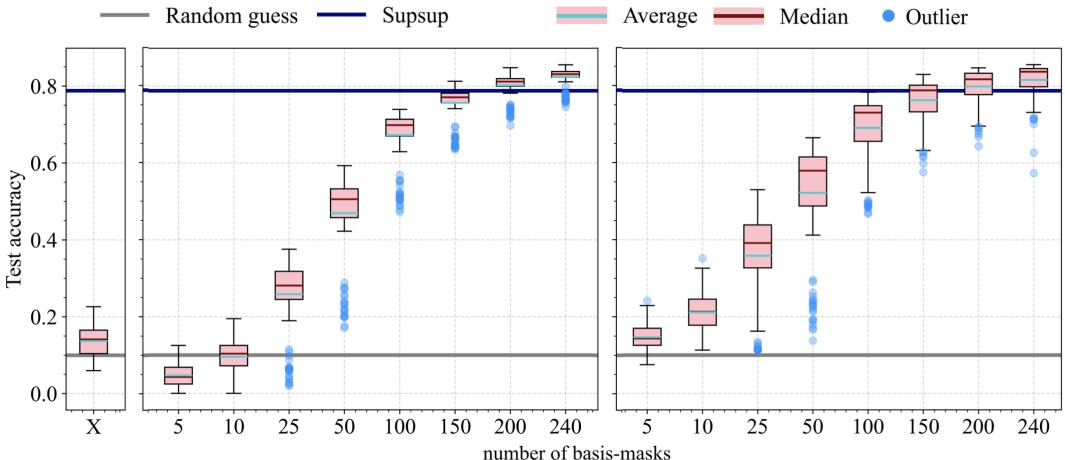

Figure 8: Average test performance (PermutedMNIST) of various strategies on 10 unseen tasks over the number of basis-masks. Boxplots from left to right: (1) A single basis-mask X, (2) ImpressLearn with random masks, (3) ImpressLearn with homogeneous masks. All results are averaged over 3 different seeds and over masks of sparsity from $\{0.95, 0.92, 0.9, 0.85, 0.8\}$

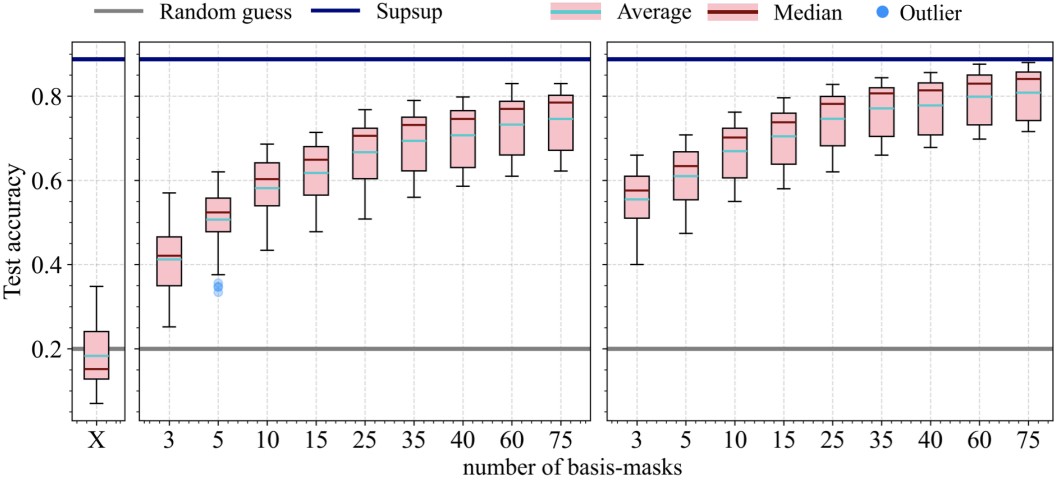

Figure 9: Average test performance (Split-CIFAR-100) of various strategies on 5 unseen tasks over the number of basis-masks. Boxplots from left to right: (1) A single basis-mask X, (2) ImpressLearn with random masks, (3) ImpressLearn with homogeneous masks. All results are averaged over 3 different seeds and over masks of sparsity from $\{0.95, 0.92, 0.9, 0.85, 0.8\}$

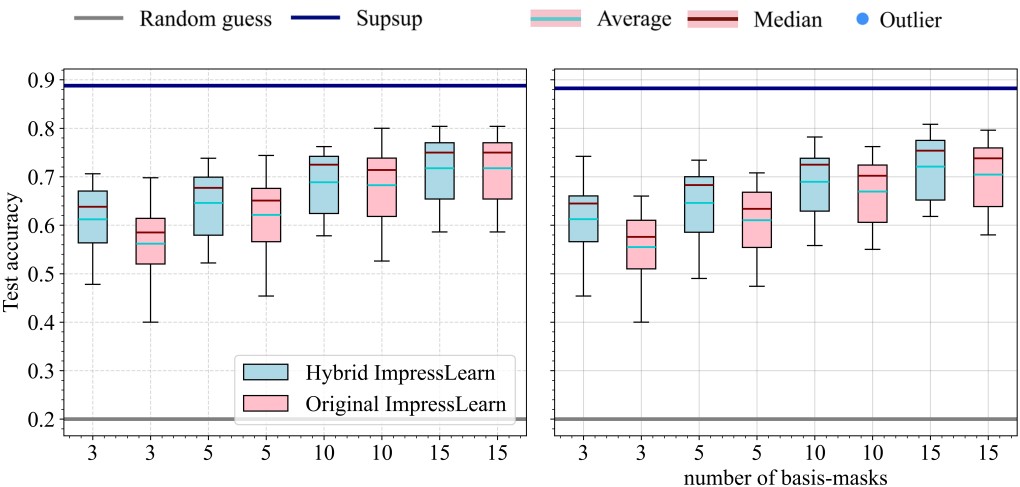

Figure 10: Average test performance (Split-CIFAR-100) of the original ImpressLearn and the hybrid ImpressLearn on 5 unseen tasks. Left boxplot: heterogeneous masks; Right boxplot: homogeneous masks. All results are averaged over 3 seeds and over masks of sparsity from $\{0.95, 0.92, 0.9, 0.85, 0.8\}$

## D   Task Order & Basis-tasks

As mentioned in Section 3, heterogeneous ImpressLearn selects the first $N$ tasks to create basis-masks. It is a reasonable approach in practice when no information about the diversity, difficulty, and nature of tasks in the stream $T$ is available. At the same time, it is fair to ask how task order will affect the performance of ImpressLearn. To address this, we investigate if there is any statistical difference between the average performance of ImpressLearn on new tasks when the task stream $T$ is shuffled using two different seeds. In particular, we run two-sample t-tests and record the corresponding p-values (Figure 11) to evaluate the performance discrepancy in different experimental setups. In addition, we print the shift in average accuracy caused by switching seeds. We observe that ImpressLearn's performance can depend on the task order, especially when basis-masks are sparse and scarce, although the direction of this change is not consistent. Interestingly, this phenomenon is not equally likely for different data streams $T$. While the performance shifts are more frequent with RotatedMNIST, they are statistically insignificant with Split-CIFAR-100. Intuitively, RotatedMNIST should be most sensitive to task order since it is more probable that considerably similar tasks (i.e., similar rotations) are adopted as basis-tasks. On the contrary, the vast number of tasks in PermutedMNIST ensures diversity of basis-tasks, and Split-CIFAR-100 features fairly independent tasks.

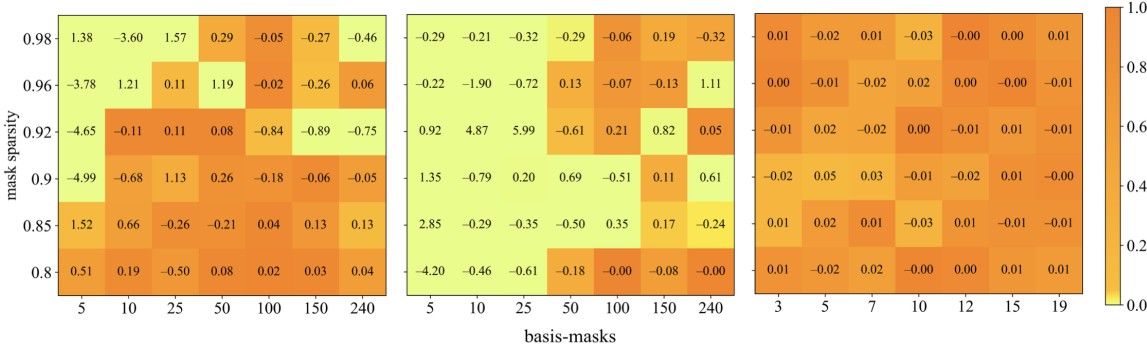

Figure 11: A heatmap of p-values corresponding to two-sample t-tests run on test accuracy of ImpressLearn under two different task sequences $T$. The numbers indicate the difference in accuracy (%) between the two sequences. Left: PermutedMNIST, Middle: RotatedMNIST, Right: Split-CIFAR-100. Yellow patches indicate p-values lower than 0.05, suggesting statistically different performance.

