# OpenReview forum: "ImpressLearn: Continual Learning via Combined Task Impressions"
_TMLR — Rejected by TMLR_

### Review · Reviewer_KmzU · 2022-11-12

**Summary Of Contributions:**

This paper proposes a new method for continual learning, where a linear combination of task-specific masks is learned to retain accuracy on previously learned tasks and achieve high accuracy on new tasks.

**Audience:**

Yes

**Broader Impact Concerns:**

Not applicable.


**Claims And Evidence:**

No

**Requested Changes:**

Clearly state the technique contribution of the methodology.

Improve the presentation of the experimental section.

**Strengths And Weaknesses:**

Strengths:

- It is an interesting insight that a linear combination is able to capture the complicated relationship between different tasks.


Weaknesses:

- It is unclear about the technical contribution of the methodology; for example, is this simply based on the utilization of the Edge-Popup training algorithm?


- The presentation of the experimental section lacks the necessary organization. For example, there is a lack of the central hypothesis of the experiments. The results are also confusing, for example, what do the circles represent in Figure 3? And why they are not in Figure 4?

---

> ### Author Response · Authors · 2022-12-30
> **Authors' response**
>
> Thank you for your review. We significantly revised the paper addressing your concerns. In particular, we regenerated all figures and added all missing legend. Further, we rewrote different parts of the paper to make our contribution, central hypothesis, and methodology clearer.
>
> The outliers (circles) are more prevalent in some plots and not as much in others. Large amounts of outliers indicate that certain supermasks were significantly underperforming, which can happen at high sparsities. It looks like this was less of a problem for ResNet-18 (which was used for Split-CIFAR-100, Figure 4) compared to LeNet-300-100 (used for PermutedMNIST, Figure 3). We hypothesize that residual connections of ResNet-18 help preserve its integrity in high sparsity regimes.
>
> Regarding the edge-popup algorithm, it is only used during basis-mask generation, while the rest of the method is original. We added a note in Section 3 stressing that any suitable mask optimization algorithm can be used in place of edge-popup, e.g., iterative magnitude pruning. Thus, while we make use of edge-popup, it is not an indispensable and by far not the most essential component of ImpressLearn.

---

### Review · Reviewer_G4os · 2022-11-18

**Summary Of Contributions:**

The authors present a method that extends prior work on learning task-specific, computationally efficient masks over network parameters for continual/multi-task learning. The method is clean and simple and demonstrates good performance at a fraction of the memory cost of the most relevant prior work.

**Audience:**

Yes

**Broader Impact Concerns:**

None that are specific to this work.

**Claims And Evidence:**

Yes

**Requested Changes:**

I would request all changes proposed in the weaknesses section above.

The most critical changes for securing my recommendation for acceptance are: Figure clarity and analysis.

The rest would simply strengthen the work.

**Strengths And Weaknesses:**

### Strengths

**************Method:************** The method is extremely simple and works well. This is a neat contribution to continual and/or multi-task learning. Especially since it matches or outperforms SupSup while being much more memory efficient.

**********Clarity:********** Generally, this paper is very well-written and intuitive.

********************************Experiments:******************************** Experiments are sufficiently thorough. I liked the experiment where SupSup was used on the last layer to see if the task-specific learning of SupSup is beneficial.

### Weaknesses

************************Clarity:************************

- Figures: The figures are a bit unclear. What are the dotted points on each of the lines? What do the orange and green dotted lines represent in the box and whisker plots?  How many values are being represented in each box plot? The choice of the green line is also strange given the SupSup baseline line is also a green dotted line. These figures should also have titles for the left and right box plots separately to make it immediately clear what regimes they’re under.
- I think the paper would be clearer if a very high-level overview of the pop-up algorithm was given in the approach section.

**************************Related work:************************** It’s also probably worth mentioning some of the older superposition-style works, e.g. [https://arxiv.org/abs/1902.05522](https://arxiv.org/abs/1902.05522)

**************************Minor Issues:**************************

- Figure quality: result figures could be exported in higher resolution; they’re a bit pixelated when looking closely. I’d recommend authors save these as PDF instead of jpg/png
- Formatting: Bolded paragraph headers look a little strange without a period after it. Maybe add periods if the authors agree.

******************Analysis:****************** I feel like some analysis is missing: notably ***why*** ImpressLearn’s $\alpha$-optimization procedure is better than entropy minimization on the different tasks. This paper could benefit from both general analysis and some (perhaps cherrypicked but illustrative) qualitative examples of incorrect predictions or learned masks across SupSup and ImpressLearn on some of their datasets, along with specific analysis of these examples.

---

> ### Author Response · Authors · 2022-12-30
> **Authors' response**
>
> Thank you for your encouraging review and feedback. We addressed your requested changes; in particular:
>
> 1. All figures are regenerated in better quality, with a clearer presentation and explicit legend.
> 2. We gave a high-level overview of the edge-popup algorithm in Section 3.
> 3. We reformatted all paragraph headers as you requested.
> 4. We added (Cheung et al., 2019) in Section 2 under the “parameter isolation” methods.
>
> Unfortunately, we were unable to run more experiments and investigate our $\alpha$-optimization in comparison with entropy minimization in these two weeks. However, we now propose this as a direction for future research in Section 5.

---

> > ### Comment · Reviewer_G4os · 2022-12-31
> > **Response to authors**
> >
> > Thanks for the response. I appreciate the much better figures and paragraph header change.
> > I have reread the entire paper, other reviews, and your responses to those reviews. I have some more comments/suggestions:
> >
> > Regarding the analysis of $\alpha$-optimization vs entropy minimization: I think it's okay to not have any experiments of comparison given a limited amount of time, but I do think it's pretty important to have this analysis and/or comparison. Could the authors at least provide some analysis or hypotheses regarding this in the experiments section even if they don't have experiments?
> >
> > Further questions:
> > - Do you need the absolute value around $\alpha_{t, i}^s$ in Eq.2? Can $\alpha$ be negative? There seems to be no restrictions on $\alpha$ in your optimization equation in Eq.1.
> >
> > Minor issues:
> > - bottom of page 1, Zhou et al. 2019 citation should be in parentheses.
> > - Section 3, header "The ImpressLearn Algorithm", "a basis masks" -> "a basis mask"

---

> > > ### Author Response · Authors · 2023-01-01
> > > **Authors' response**
> > >
> > > Thanks for your prompt reply and for taking time to reread the revised manuscript.  In this version, we partially rewrote the paragraph titled "GN regime" in Section 4, stating our explanation of why ImpressLearn might work better than SupSup. In particular, we now write:
> > >
> > > "...For most architecture-dataset pairs, our strategy either determines the correct basis-mask or discovers an even better-performing linear combination of basis-masks (the right-most plots in Figures 2, 3, 4, and 5), facilitating positive knowledge transfer to previous tasks. On the other hand, SupSup’s one-shot procedure outputs only a single binary mask and will retain the original performance at best...".
> > >
> > > Thus, we believe that it is not the optimization procedure itself but rather allowing the output to be a combination of masks instead of just a single binary mask.
> > >
> > > The reason to have the absolute value around $\alpha_{t,i}^s$ is to have $\ell_1$ regularization in an attempt to obtain a sparse solution (see the paragraph above equation 2). The $\alpha$-values are unconstrained and can be negative.
> > >
> > > Thank you for pointing out other minor issues and typos---we fixed that.

---

> > > > ### Comment · Reviewer_G4os · 2023-01-02
> > > > **Further comments**
> > > >
> > > > Thanks for the prompt response, I appreciate the extra sentences you added below "GN regime" and I think it makes the experiments section more complete now.
> > > >
> > > > > The reason to have the absolute value around $\alpha$ is to have regularization in an attempt to obtain a sparse solution (see the paragraph above equation 2). The $\alpha$-values are unconstrained and can be negative.
> > > >
> > > > Ah I see, I originally believed that the $\alpha$ values were a linear combination that summed to 1. Then in this case, I believe the loss that you're applying doesn't actually encourage a sparse solution, nor is it equivalent to $l_1$ regularization. It actually encourages all $\alpha$ values to be 1 to minimize this part of the loss, right (also using squared loss which wouldn't be $l_1$ either)? Is this a mistake in the equation or is this the actual optimization objective used for the experiments (or my misunderstanding)?

---

> > > > > ### Author Response · Authors · 2023-01-02
> > > > > **Authors' response**
> > > > >
> > > > > The equation (2) is correct and is the actual optimization objective of ImpressLearn. For each layer $i\in[d]$, it encourages the sum $\sum_{t=1}^N|\alpha^s_{t,i}|$ of the absolute values of coefficients to sum to $1$, which is essentially $\ell_1$ regularization but just not around $0$ but rather around $1$. In fact, plotting the level curves of the regularization term in equation (2), we see a piecewise-linear graph with "Lasso-like corners" that encourage a sparse solution. [Here we plot](https://www.desmos.com/calculator/ggbik8mudq), $(|x|+|y|-1)^2=C$ for different values of the constant $C$ as an example.

---

> > > > > > ### Comment · Reviewer_G4os · 2023-01-02
> > > > > > **Final response**
> > > > > >
> > > > > > All my concerns have been addressed. I will be submitting my recommendation now. Thank you!

---

### Review · Reviewer_qohJ · 2022-12-17

**Summary Of Contributions:**

This paper proposes the ImpressLearn model for continual learning. The proposed method first obtains a set of basis masks over the randomly initialized backbone model with a single task (homogeneous setting) or multiple tasks (heterogeneous setting). When given new tasks, a linear combination of the basis masks is obtained via optimization and used to perform the new tasks.

The main contributions:

1. Experimentally demonstrate the effectiveness of the strategy to perform new tasks with linear combinations of learnt basis masks overlaying a randomly initialized backbone.

2. The proposed method is more memory efficient than an existing similar method, SupSup, when the number of tasks is large.

**Audience:**

Yes

**Broader Impact Concerns:**

No other concerns

**Claims And Evidence:**

No

**Requested Changes:**

Some confusing parts are listed below.

1. How to choose the tasks for obtaining the basis masks?

2. The symbols in Figure 2 requires further explanations, for examples, what are the orange line and green dashed line within each small box in the figure? What are the small circles?

3. In Example: LeNet-300-100 on RotMNIST, what does 3x10=30 parameters mean? where does the multiplier 3 come from?

4. It would be better to show the statistics including the number of tasks, number of classes per task with a table (can be put in Appendix).



**Strengths And Weaknesses:**

Strengths:

1. Experimentally demonstrate the effectiveness of the strategy to perform new tasks with linear combinations of learnt basis masks overlaying a randomly initialized backbone.

2. The proposed method is more memory efficient than an existing similar method, SupSup, when the number of tasks is large.

Weaknesses:

1. For continual learning, it is important to see how the learnt model can perform on previous tasks, but in the experiments (figure 3,4,5), it seems that the testing is only conducted on new tasks while the performance on previous tasks is unknown.
Besides, the proposed ImpressLearn requires pre-training to obtain the basis masks, which is not required by the baselines. In this case, how to ensure a fair training strategy? E.g., which part of the dataset is used for obtaining the basis masks, and how would the baselines use this part of dataset?

2. In GN scenario, the proposed method has to optimize the linear coefficients to find a proper combination of the basis masks. This solution has actually been considered by SupSup, but was not adopted because of the high computation burden incured by the linear coefficient optimization. Therefore, this strategy, which is a major part of the proposed method, can hardly be considered as a contribution, but actually has certain limitation.

3. The result that homogeneous masks perform worse than SupSup implies that the construction of the basis masks is crucial, therefore, how to properly choose the tasks for obtaining the basis masks may be a chellenge in practice, especially when the new tasks are unknown.

4. When calculating the parameters to store, it is unfair to only count the linear coefficients and ignore the basis masks, which is the major space consumption. In the Example: LeNet-300-100 on RotMNIST, since there are only 10 tasks, SupSup only has to store 10 masks. However, in different configurations shown in figure 2, ImpressLearn could store from 5 to 240 basis masks. When ImpressLearn stores 10 basis, the total space consumption of the masks is same as SupSup but the linear coefficients of ImpressLearn requires extra space. When the number of basis masks exceeds the number of tasks, the space consumption is actually much more than SupSup.

---

> ### Author Response · Authors · 2022-12-30
> **Authors' response**
>
> Thank you for the detailed review. We reply to your questions below:
>
> Weaknesses---
>
> 1. We do report performance on the learned tasks. In our setup, previous tasks can be split into two types. First, these are basis-tasks that have their own binary mask computed using edge-popup algorithm. The performance on these tasks is presented in the rightmost plots in Figures 3, 4, and 5. Second, there are previously unseen tasks that receive their parameter mask by linearly combining basis-masks in the $\alpha$-optimization procedure. The performance on these tasks is recorded in the middle plots of Figures 3, 4, and 5 when these tasks are treated as ``new'' Clearly, the performance on these tasks will remain the same after subsequent learning because basis-masks and associated coefficients will remain intact in memory and can be recovered using task's identity or rediscovered using $\alpha$-optimization in the GN regime.
>
> 2. After carefully reviewing the paper (Wortsman et al., 2020), we could not find them mentioning optimizing for a *combination* of masks; rather, they propose several algorithms ("one-shot" and "binary") that output a *single* mask. Additionally, the computational complexity in their case may come from the necessity of optimizing over masks associated with all previous tasks, whereas our optimization in the GN inference regime is over basis-mask only.
>
> 3. We think it is natural to choose one basis-mask per each of the first $N$ seen tasks since there is no information about data distribution of future tasks. $N$ here is the number of basis-tasks and is left as a hyperparameter, controlling the performance-efficiency trade-off. We report results over various different values of $N$.
>
> 4. It is true that ImpressLearn must store binary masks and, in scenarios where the number of tasks in the stream $T$ is small, may end up taking more memory than SupSup to match its performance. We are aware of this shortcoming and stated it in the limitations section of the paper. Throughout the paper, we argue that ImpressLearn is most suitable for task streams with a large number of tasks (e.g., PermutedMNIST or RotatedMNIST) so that its potential memory savings can be realized. That said, we want to emphasize that we are explicit about the amounts of memory required by ImpressLearn. For example, the per-task memory requirements reported in Table 1 are computed by amortizing the cost of storing the specified number of basis-masks over all possible tasks, as stated in the caption.
>
> Requested changes---
>
> 1. Unfortunately, we do not have a thorough answer for that. As mentioned in the paper, we take the first $N$ tasks as basis-tasks as a reasonable heuristic provided that not much information is available beforehand.
>
> 2. All figures are regenerated with a clearer presentation and explicit legend.
>
> 3. During learning of a new task, ImpressLearn allocates a single floating point parameter per basis-task per layer of the network, so 3 here is the number of layers in LeNet-300-100. We added a few words to the paper to clarify this.
>
> 4. In the revised version, we put these statistics in Table 1.

---

> > ### Comment · Reviewer_qohJ · 2023-01-13
> > **Thanks for the detailed explanations from the authors**
> >
> > After reading the responses, some of the concerns are resolved, but some detected weakness still exist. Details are listed below.
> >
> > 1. The problem regarding the performance on previously learnt tasks is resolved.
> >
> > 2. In the paragraphs before and after Equation 3 in page 5, the authors stated that gradient descent on the cosfficients is one option, but is not adopted because they want an optimization method with fixed sub-linear run time.
> >
> > 3. As explained by the authors, there is no strategy to ensure the diversity of the basis tasks, the proposed method only select the first N tasks. When the first N tasks are very similar and the following tasks are much more different, there is no mechanism to replace the old masks with new ones. Therefore, I still regard this as a weakness.
> >
> > 4. The authors agree with this limitation, therefore the memory consumption is actually not an advantage.
> >
> > The requested changes are well addressed.

---

> > > ### Author Response · Authors · 2023-01-14
> > > **Authors' Response**
> > >
> > > Thank you for your reply; we would like to provide further clarifications and stress a few points:
> > >
> > > 2. Regardless of the optimization method, Wortsman et al. (2020) use the final coefficients to identify a *single* binary mask that pertains to the input task. On the other hand, we optimize for a *combination* of masks, which frequently outperforms a dedicated binary mask trained for that particular task due to positive backwards knowledge transfer not possible with SupSup. Moreover, our GN optimization is over a constant (hence, sublinear) number of basis-masks by default, although we also tried optimizing over both binary basis-masks and previously computed real-valued masks for seen non-basis-tasks.
> > >
> > > 3. ImpressLearn can be trivially extended to account for a potentially shifting task distribution: given an inadequate validation performance on a new task, a new binary mask can be generated to replace one of the existing ones. The specific task selection and mask replacement criteria are subject for an extensive ablation study, which was out of the paper's scope but would be an interesting direction for future research.
> > >
> > > 4. We disagree. As we stressed both in the paper and our first response, memory savings depend on the nature of the task stream and, in particular, the number of potential tasks. We described scenarios where using ImpressLearn yields considerable advantages, and, hence, it is incorrect to say that it does not bring any.

---

### Review · Reviewer_Byhw · 2022-12-18

**Summary Of Contributions:**

This paper extends the SupSup algorithm for continual learning. Instead of learning a new mask for each new task, it proposes to optimize a linear combination of basis masks, either heterogeneous or homogeneous. Experiments are conducted on three datasets under both the standard setting and the GN setting.

**Audience:**

Yes

**Claims And Evidence:**

No

**Requested Changes:**

Please address above weaknesses, especially Weakness 2 and Weakness 3.

**Strengths And Weaknesses:**

Strengths:

1. It is interesting to see a simple linear combination of basis masks could achieve some preliminary positive results for continual learning.

2. The proposed method has better storage efficiency compared to the baseline.

3. The potential application to protect against data drifts with flexible adjustments discussed on page 9 is interesting.

Weaknesses:

1. Although good experimental results are obtained on MNIST, the proposed method shows inferior performance on CIFAR and ImageNet compared to the baseline. No enough evidence is provided on the generalization ability of this method on large and complex datasets.

2. How are the basis tasks selected and the impact of different basis tasks are unknown. Will the similarity between basis tasks and new tasks be a crucial factor for the performance or not?

3. In Figure 2, what are the yellow solid line and the green dotted line? Are they average value or median value? And why are there many outliers with pretty low accuracies? What is the reason for this?

---

> ### Author Response · Authors · 2022-12-30
> **Authors' response**
>
> Thank you for your feedback and suggestions. We would like to address your comments below:
>
> 1. We agree that ImpressLearn, in its current state and the number of basis-masks obtained, was unable to match the performance of SupSup on Split-CIFAR-100, and Split-ImageNet, which we explicitly state in Section 4 as well as in the Limitations (Section 5). At the same time, we want to emphasize that these benchmarks are not ideal use-cases for ImpressLearn as the number of tasks is small. As such, it does not allow to generate enough heterogeneous basis-masks, which may partly explain the suboptimal performance.
>
> 2. Unfortunately, we have not investigated this in our original study and two weeks was not enough for us to do it now. The said, the ordering of tasks was seeded, so it has been factored out from the presented results.
>
> 3. We modified our figures substantially and made all legend explicit. You are right, these lines denote the mean and the median. As per the outliers, many of those come from underperforming high-sparsity masks (e.g., 98% sparse networks).

---

> > ### Comment · Reviewer_Byhw · 2023-01-02
> > **Response to authors**
> >
> > Thank you very much for the response and revision. As for the Weakness 2, it would be very helpful to address my concern if some analysis  can be provided about the impact of different basis tasks. For example, to what extent a different seed will affect the results?

---

> > > ### Author Response · Authors · 2023-01-03
> > > **Authors' response**
> > >
> > > Please check Appendix 4 of a new revision that we just added.

---

> > > > ### Comment · Reviewer_Byhw · 2023-01-05
> > > > **Response to authors**
> > > >
> > > > Thank you very much for the update. It is interesting to see this statistical analysis. However, in Appendix 4, it is unclear in which direction a different random seed affect the results. It would be more meaningful to have a comparison showing in what way a different seed would change test performance reported in paper (Figure 2, 3, 4). Would the results be higher, lower or similar?

---

> > > > > ### Author Response · Authors · 2023-01-05
> > > > > **Authors' response**
> > > > >
> > > > > Thank you for the clarification. We uploaded a new revision with heatmaps in Appendix 4 now showing the difference in average performance when switching between the two seeds. As expected, the magnitude of the change is correlated with the corresponding p-value (indicating statistically significant change), but the direction of this change is not consistent. The data used for Figure 11 is the same data that generated Figures 2, 3, and 4.

---

### Author Response · Authors · 2022-12-29
**Replies and revisions pending.**

We thank all reviewers for their valuable feedback and proposed changes. Sorry for the delayed response---we were actively working on the suggested edits. We will reply to all reviews by EOD, Dec. 30th, as well as resubmit the modified manuscript.

---

### Decision · Action_Editors · 2023-01-23

**Recommendation:** Reject

**Comment:**

This paper studied masking schemes for continual learning. Given a set of basis masks, a linear combination is optimised to derive the mask for new tasks. The reviewers agreed that part of their major concerns has been well addressed by the authors' response. But two reviewers still were not fully satisfied and were leaning reject. In particular, there is no strategy to ensure the diversity of the basis tasks/masks, which may lead to problems given the uncertain new tasks in continual learning. Also, the experimental results on larger and more complex datasets are not significant enough and memory consumption seems to be a disadvantage that needs extra attention.

Before considering a resubmission, the authors are strongly suggested to take major efforts to redevelop their algorithm to fix the reviewers' concerns about the techniques (e.g., diversity issue) and demonstrate more clear experimental advantages over SupSup (as SupSup is the authors' major baseline).

**Audience:**

The proposed algorithm strongly relies on an existing continual learning method SupSup, which makes the algorithm (and its insight) hard to generalise and help more researchers in the broad continual learning field. The authors pointed out the flaws of SupSup, which is a nice observation in this paper, but the current experimental results are hard to convince the reviewers/readers about the usefulness of the proposed algorithm. Thus, the reviewers and I are afraid that very few audiences could be interested in the proposed improvements.

**Claims And Evidence:**

This paper criticises that an existing SupSup method restricts knowledge transfer between tasks and requires considerable memory overhead for each additional task. Based on SupSup, the authors develop a new algorithm ImpressLearn that uses a linear combination to derive the mask for new tasks.
The motivation of this paper is clear and the authors have developed a reasonable algorithm to improve SupSup. But it lacks solid evidence to support the effectiveness of the proposed algorithm, especially considering the algorithm's inferior results on larger and more complex datasets. Though the authors gave some explanations on why the proposed algorithm did not work very well, it makes the authors' criticism of SupSup untenable, as the new algorithm is also a bit fragile and cannot well fix SupSup.